# Rape Myths and Verdict Systems: What Is Influencing Conviction Rates in Rape Trials in Scotland?

**DOI:** 10.3390/bs14070619

**Published:** 2024-07-21

**Authors:** Lee John Curley, Martin Lages, Pamela J. Sime, James Munro

**Affiliations:** 1Department of Psychology, Glasgow Caledonian University, Glasgow G4 0BA, UK; lee.curley@gcu.ac.uk (L.J.C.); pamela.ritchie@gcu.ac.uk (P.J.S.); 2School of Psychology & Neuroscience, University of Glasgow, Glasgow G12 8QQ, UK; martin.lages@glasgow.ac.uk; 3Faculty of Arts and Social Sciences, School of Psychology and Counselling, The Open University, Milton Keynes MK7 6AA, UK

**Keywords:** rape myths, juror decision-making, not proven verdict, three-verdict system, Scottish legal system, jury reform

## Abstract

The Scottish verdict system includes three verdicts: ‘guilty’, ‘not guilty’ and ‘not proven’. Politicians propose that the three-verdict system is partially to blame for the low conviction rate of rape, whereas research suggests that rape myths may be having a larger impact. To test the effects of varying verdict systems (guilty, not guilty and not proven; guilty and not guilty; a series of proven and not proven verdicts) and rape myths on juror verdicts. A total of 180 participants answered questions regarding their acceptance of rape myths using the Acceptance of Modern Myth and Sexual Aggression (AMMSA) scale. They then watched a staged rape trial filmed in a real courtroom and reached a verdict. Participants also provided longer-form answers on which thematical analysis was conducted. The main findings are as follows: (1) The special verdict system leads to a higher conviction rate than the other systems when rape myth acceptance is controlled for. (2) The higher the rape myth acceptance, the more favourably the accused was perceived and the less favourably the complainer was perceived.

## 1. Introduction

### 1.1. Context

Scotland is, as of 2024, a unique legal jurisdiction when compared to other parts of the world (i.e., Australia, the United States of America (USA), and England and Wales). Scotland, unlike the other countries mentioned, has three verdicts available to jurors: guilty, not guilty, and not proven [1]. The not proven verdict is an acquittal verdict, just like the not guilty verdict, with accused individuals being free to leave once they have received said verdict. The not proven verdict is not legally defined [2] and has caused a great deal of debate and discussion in Scotland in recent years due to a perception of its overuse in crimes such as rape and sexual assault [3]. Other unique features of the Scottish jury system include the following: 15-person juries; a simple majority verdict rule (8 out 15 jurors are enough for a decisive verdict); and corroboration (two pieces of credible and reliable information are needed to support the claim that “an offence was committed”) [4].

These unique elements have come under scrutiny in recent years from organisations such as Rape Crisis Scotland and politicians, who propose that some of these factors (particularly the not proven verdict) are the reason for the low conviction rate of rape and sexual assault in Scotland [3,5,6,7]. They believe this because the not proven verdict is used frequently in trials involving serious crimes such as rape and murder [5,6,7]. Due to this critique, following consultations with experts and laypeople, as well as the Scottish Jury Study [1], the Scottish Government has proposed the “Victims, Witnesses and Justice Reform” (Scotland) Bill (2023) [8]. This bill suggests the following reforms:Abolish the controversial ‘not proven’ verdict;Change the current jury number from 15 to 12;Change the majority rule from 8/15 to 8/12;Conduct a pilot of rape trials where the final verdict is reached by a single judge with no jury [8].

It is hoped by the Scottish Government that these changes will increase convictions in rape and sexual assault trials in Scotland. This want of the Scottish Government is not based on strong academic foundations; only a very small number of studies have been completed on which they might base their decisions. 

Research from Ormston et al. (2019) [1], which was the largest jury study that has ever been conducted in the UK, found that jury verdict system (the Scottish thee-verdict system versus the Anglo-American two-verdict system) and jury size (12 vs. 15) did not have any significant effects on the likelihood of a juror reaching a guilty verdict in a rape trial, but it did have a significant effect in a physical assault trial (with those in the two-verdict system and on a 15-person jury being more likely to give a guilty verdict). Further, jurors using a simple majority verdict rule were more likely to reach a guilty verdict than jurors using a unanimous verdict rule. Currently, there is no research that suggests that the planned changes to the verdict system and jury size will affect jury decisions, however, as all mentioned effects were at the juror level. 

Based on the current, and limited, research, we suggest that changes from a 15-person to a 12-person jury and/or from a three-verdict system to a two-verdict system are unlikely to influence conviction rates significantly in rape and sexual assault trials, despite the availability of the not proven verdict influencing other crime types such as homicide and physical assault [1,2,9,10]. However, the removal of the simple majority verdict rule is likely to have a negative effect on conviction rates in rape trials [1]. Issues regarding the pilot of judge-only trials have been the topic of conversation and discussion by academics, politicians, and legal professionals, with many commenting on the fact that experts can be and often are biased [11,12,13]. There is currently no robust academic basis for the suggestion that judge-only trials would increase conviction rates for rape trials. However, there is a wide literature base showing the negative effect that rape myths have on juror decision-making [14].

Rape myths are false or baseless misconceptions about sexual assault or rape that exist in and are perpetuated by society and its media [15]. There are many different rape myths, including the idea that some characteristic or decision of the victim provoked the assault (rather than the perpetrator being responsible [16]) and that false reporting of rape is common [17]. These myths contribute to victim-blaming and often lead to the victim experiencing stigma, disbelief, and a lack of regard for the seriousness of the crime that they have experienced [18]. Rape myths are actively invoked by defence lawyers in rape trials [19] and are likely to play a significant role in juror and jury decision-making. Rape myths may explain low conviction rates in rape trials in Scotland and may overshadow potential effects of the verdict system.

### 1.2. Background

There is a lot of scepticism surrounding the not proven verdict due to the randomness of the three-verdict system’s origins and its high usage in sex-related trials [1,2,20]. For example, Rape Crisis Scotland and Miss M have campaigned for the removal of the not proven verdict, placing the blame for the low conviction rates in rape trials on the availability of the verdict.

Scotland’s current three-verdict system of guilty, not guilty, and not proven originated by accident rather than by evidence-based research or reasoning [21,22]. Originally, in Scotland, juries could give ‘general verdicts’ regarding the guilt of the accused [21]. General verdicts allow juries to make a general judgement of the accused regarding whether they found them guilty or not guilty, without listing specific points which they believed were found proven or not proven [21]. During this time, though, there was a lack of consistency regarding terms for guilt and innocence. Some declared guilt through Latin (‘convictus’), whereas others declared guilt through Scots (‘had done wrangis’; Jackson et al. [23]). Likewise, terms such as ‘made qwyt’ and/or ‘clene and sakles’ (Scots) were also used to declare an acquittal [21,23].

In the late 1600s, however, the power of the jury was diminished, and they were no longer allowed to give general verdicts and, thus, make decisions regarding the guilt of the accused [21]. Rather, they were instead asked to give special verdicts. A special verdict is when a jury finds the facts only, leaving the judgement to the judge/court [23]. In this system, juries are asked to make decisions regarding whether specific facts were proven or not proven. For example, in a modern rape trial, a jury might be tasked in such a system with making special verdicts regarding whether the complainer consented and/or the accused knew that they did not have consent. The judge would then make a general verdict (guilty or not guilty) based on these special verdicts (proven and not proven). By the early 1700s, juries were once again able to give general verdicts, this time with the not proven verdict also being used as a general verdict, in addition to the guilty and not guilty verdicts [23,24].

Despite the uniqueness of the Scottish verdict- system and its recent critique, experimental research exploring the effect of a three-verdict system on juror verdicts is relatively recent, with the first paper being published in 2007 [10]. A recent meta-analysis by Jackson et al. (2024) [23] used all existing mock-trial data to that date for investigating the two versus the three-verdict system from 10 studies and 1778 mock jurors. They found that the chances of a juror convicting in a three-verdict system are approximately 10% lower than a juror convicting in a two-verdict system. Jurors reach different verdicts depending on the verdicts available to them. Due to the limited amount of data, however, no researcher has been able to investigate if the verdict system interacted with other factors such as crime type [23]. Only recently was research published exploring the potential influence of the verdict system on jurors’ decisions regarding a mock retrial of a crime. Munro et al. (2024) found that significantly more acquittals were given by jurors when they were told it was a retrial, regardless of whether they were told that the original trial resulted in a not proven or not guilty acquittal verdict [25].

Nevertheless, two papers included in Jackson et al.’s (2024) meta-analysis [23] may be able to illuminate if any interaction may exist between the verdict system and crime type at the juror level. First, Hope at al. (2008) [2] found that the availability of the not proven verdict did not influence the number of guilty verdicts reached by jurors in a sexual assault trial but did have a significant effect in a physical assault trial. Second, Ormston et al. (2019) [1] found a significant effect of verdict system (two versus three) in a physical assault trial and not a rape trial. Though caution should be used when drawing conclusions about such a far-reaching phenomenon based on only two studies, it appears that the availability of the not proven verdict may only influence juror decision-making in trials not involving rape and sexual assault (e.g., homicide and physical assault). One explanation for this may be that rape myths are causing floor effects in mock juror studies, meaning that as the conviction rate is so low, it is difficult to establish if other factors (such as different verdict systems) also influence juror verdicts [9]. However, if rape myths are controlled for (by juror selection or statistical modelling), then differences in convictions may appear between varying verdict systems.

Rape myths are negative and false beliefs that have been shown, through experimental research, to have an influence on how jurors view the evidence and legal actors (complainer and accused) in a rape trial [26,27]. They are a class of beliefs or attitudes towards relationships, abuse, and sex that contribute towards the believability or credibility of the accused and complainers, similar to attitudes towards sex trafficking [28]. Rape myths have been shown empirically to influence jurors in other jurisdictions (e.g., England and Wales, the USA, and South Africa [27,29,30], and research from Chalmers, Leverick, and Munro (2021) [31] has found evidence for rape myths being present in deliberations in Scottish juries. It is therefore likely that rape myths also influence verdict choice in Scotland. Provision for rape victims/survivors is a pressing and ongoing matter among the Scottish Government, researchers, and, of course, campaigns led by survivors. To illustrate the saliency of the issue, a work by Richardson (2024) [32] evaluates one pilot scheme by the government that made high court transcripts available to complainers of rape.

Rape myths are commonly measured using rape myth questionnaires such as the Acceptance of Modern Myths about Sexual Aggression scale (AMMSA [33]). The AMMSA scale is a 30-item measure where participants state how much they agree or disagree with statements on a seven-point scale [33]; the AMMSA scale and its items will be explained further in the Materials and Methods Section. These statements focus on rape myths surrounding consent, sex, and the behaviour of men and women. Lilley, Willmott, Mojtahedi, and Labhardt (2023) [34] go into great depth regarding the research demonstrating the factual inaccuracy of key rape myths and the ways in which any endorsement of such rape myths by legal actors can inhibit pathways to justice and create barriers to reporting rape. Research has consistently shown that men are more likely to advocate rape myths [35] and that rape myth measures can be used to predict verdicts in mock rape trials [14,36]. Furthermore, rape myths may be decreasing conviction rates in Scottish mock trials, like they do in other jurisdictions, making it difficult to assess if verdict system effects exist in experimental trials. Previous research might demonstrate a similar effect, in which gender differences in attitude towards rape, in the judgement of rape cases, might occlude other effects such as the presence of high-functioning ASD amongst the complainers or accused [37].

In addition to the common suggestion that the not proven verdict should be abolished, there are some who suggest that a more purposeful reform would be to change to a binary verdict system of proven and not proven. For example, Curley et al. (2021) [38] asked legal professionals to rank three different verdict systems: (1) guilty, not guilty, and not proven; (2) guilty and not guilty; (3) proven and not proven. These legal professionals ranked proven and not proven as their most preferred system; with guilty and not guilty being the least preferred system. One reason for this is that a proven and not proven system may help jurors to adopt their supposed role, i.e., to focus on whether the Crown has proven their case beyond reasonable doubt [3,38].

Curley et al. (2022) [3] investigated the effect that the three different verdict systems mentioned above had on juror decision-making in a homicide trial. Here, the proven and not proven verdict system was used as a general verdict system, in the same way as guilty and not guilty. They found that the proven and not proven verdict system led to a similar number of convictions as the current three-verdict system of guilty, not guilty, and not proven. Further, they found that both the proven and not proven and three-verdict systems led to significantly fewer guilty verdicts than the two-verdict system of guilty and not guilty. These findings demonstrate that general verdicts which focus on proof (i.e., those containing the word ‘proven’) may cause jurors to use rational decision processing, which may then decrease the chance of a conviction in non-sex-related crime types [3]. This may be because the focus on proof may stop jurors from making decisions based on morality and punitive action, thus attenuating any pro-prosecution bias they may have, and instead focus them on the claims of the Crown and its evidence. Previous research has even shown that guilty verdicts are much more likely to be chosen in homicide trials when non-rational decision strategies are employed [24,39,40]. This effect may be different in sex-related crimes, however, as rape myths may promote a defence-based bias. Therefore, any promotion of rational thought which attenuates the role of rape myths is likely to increase convictions.

No previous research has investigated the impact that a special verdict system of proven and not proven may have on conviction rates in rape trials when compared to other verdict systems. The special verdict system here would be a system similar to the one that existed in Scotland in the late 17th century [23]. Asking jurors to focus on specific facts of a case (i.e., did the accused penetrate the complainer? Was consent given? Did the accused know they did not have consent?) may help to focus these jurors on the evidence provided and whether claims against the accused are proven. By priming jurors to focus on the proof provided (i.e., the claims against the accused) and by breaking down the final decision into smaller, more manageable decisions, it may increase the elaboration likelihood of the jurors (the probability that jurors will consider in detail the content of the case [41]). This would thus give them more cognitive capacity to make rational decisions not governed by bias (i.e., rape myths), and conviction rates may increase as a result. Research from decision science has shown that prompts towards rationality such as “think like a scientist” have promoted more rational decision-making [42]. Verdicts such as proven and not proven may have a similar effect. From these special verdicts, the researchers will then make a general verdict of guilty and not guilty in the same way as a judge would in a real-world trial using special verdicts.

In summary, the current study investigates the effects of three different verdict systems ((1) guilty, not guilty and not proven; (2) guilty and Not guilty; (3) special verdicts of proven and not proven) and rape myths on juror verdicts. The hypotheses are as follows:(1)The AMMSA score will predict convictions, with the higher the rape myth score, the fewer convictions there will be. This has been confirmed in other jurisdictions such as England and Wales and the USA [14].(2)When controlling for the AMMSA score, the special verdict system (proven and not proven) will lead to more convictions than either of the general verdict systems (the Anglo-American two-verdict system and Scottish three-verdict system). This is because the system may increase the elaboration likelihood and prime rational decision-making, thus decreasing the impacts of biases that promote acquittal (i.e., rape myths [3]).(3)When controlling for the AMMSA score, there will be more convictions in the Anglo-American verdict system when compared to the Scottish verdict system. This is because rape myths may have caused floor effects in previous studies [1,2], and when controlled for, there may be more convictions in the two-verdict system, similar to what has been found for other crime types [1,2,3].

## 2. Materials and Methods

### 2.1. Design

We utilised an experimental design, with verdict system as a between-subjects factor and Acceptance of Modern Myth and Sexual Aggression (AMMSA; [43]) score as a covariate. The between-subjects factor had three levels: (1) a guilty, not guilty, and not proven verdict system; (2) a guilty and not guilty verdict system; (3) a series of 3 proven and not proven ‘special’ verdicts. We will refer to these as the ‘Scottish’, ‘Anglo-American’, and ‘Special’ verdict systems, respectively, when discussing our methods and results. Our key dependent measure, which was taken using an online questionnaire tool, was verdict—the labels for conviction and acquittal differed between groups. We also asked participants to self-report their beliefs about the accused and victim regarding their credibility, reliability, professionalness, education, likeability, trustworthiness, confidence in their own testimony, accuracy and likelihood of trying to deceive the jury, as well as how convincing they were and how easily the juror could be convinced to change their opinion about them. Conviction and acquittal rates were then calculated from individual jurors’ verdicts; see Data Preparation for more detail. These measures, additional questions, and the stimulus video are detailed in the section below. This study received ethical approval from [redacted for review]. British Psychological Society guidelines were followed. Our key ethical considerations included collecting, disseminating and storing anonymised data securely; withholding from participants only the particular variables that were being studied in the information sheet and providing this information in the debrief; and identifying that participants may have become upset if the video trial or the topic and associated contents reminded them of a negative experience. We provided contact details for the research team, an unrelated third party, and the Samaritans.

### 2.2. Participants

The following exclusion and inclusion criteria were used to correspond to the requirements for jury eligibility in Scotland: (1) participants had to be over 18 years old; (2) participants had to be registered as a parliamentary or local government elector in the United Kingdom (i.e., on the electoral roll); and (3) participants must have lived in the United Kingdom, Channel Islands, or Isle of Man for any period of at least 5 years since they were 13 years old. In the data collection phase of the study, 180 eligible participants were recruited using the online platform Prolific. The mean age of the participants was 42.70 years (SD = 13.56; min = 19; max = 91), and the sample consisted of 85 females, 90 males, three non-binary people, one person who stated that they were another gender, and one person who stated that they preferred not to say. In total, 180 participants stated they were born in and/or lived in Scotland or stated they were based in Scotland for the purposes of our screening on Prolific. Moreover, 178 participants stated that they were registered to vote, with 2 stating they were not registered to vote; these individuals were included in data analysis as it was deemed that there would be no difference in their decision-making.

In addition, the sample was predominantly Caucasian: 169 participants reported their ethnicity as white British, 4 as Chinese, 3 as Asian Pakistani and 1 each as Other White, Asian Indian, Mixed White and Black Caribbean, and Mixed White and Black African. Or sample roughly matched the demographics of Scotland, with 93.88% being White Scottish/British and 91.8% of the country being White Scottish/British (Scotland Census [20]). In total, 104 participants stated they had been called for jury service in Scotland before, with 32 stating that they served on a jury in Scotland, and 8 of our participants had studied Scots law.

### 2.3. Video Trial

The mock trial used in the current study was developed by Ipsos MORI Scotland (London, UK) for research commissioned by the Scottish Government in 2019 [1]. The video documents a rape trial in which current trial practices were used. The trials were scripted by members of the Scottish Government’s Research Advisory Group, which included a judge, a sheriff, an advocate, and a solicitor. The video allowed participants to watch a realistic mock trial, filmed in a real courtroom, with the use of a retired judge and actors playing the roles of witnesses and legal professionals. The video lasted approximately 68 min. Edits were made to the contents of the judge’s instructions and closing statements by the defence and prosecution to cut any mention of the verdicts available to the jury—since available verdicts was the experimental factor we wished to manipulate.

There were three post-video verdict directions to the jury, which reflect our experimental conditions: (1) Scottish: guilty, not guilty, and not proven; (2) Anglo-American: guilty and not guilty; (3) Special: a series of three proven and not proven special verdicts. In the video, the jury is instructed by the judge, several witnesses present evidence, and the jury hears closing statements from each of the respective counsels.

### 2.4. Questionnaires

There were five main parts to the questionnaire: demographics; the AMMSA scale; verdict decision; perceptions of the accused and the victim; and post-trial questions. In the demographics part, participants were asked questions relating to their ethnicity, gender, and age.

The AMMSA scale [43] is a series of 30 statements that can be answered on a Likert scale of 1 (completely disagree) to 7 (completely agree). The statements pose opinions about rape myths, such as ***“Once a man and woman have started “making out”, a women’s misgivings against sex will automatically disappear’*** and ***‘It is a biological necessity for men to release sexual pressure from time to time”***. The minimum possible score on the AMMSA scale is 30, and the maximum is 210. There are several instruments for measuring an individual’s belief or acceptance of rape myths. Two of the most common are the Illinois Rape Myth Acceptance Scale (IRMAS [44]) and the AMMSA scale. Leverick (2020) [26] found in her review that each scale is successful at detecting individual differences and described the AMMSA scale as ‘One of the most up-to-date RMA [Rape Myth Acceptance] scales’. Thus, this scale was selected for use here.

In the verdict decision section, participants were asked to give a verdict regarding the evidence presented in the trial video. Participants were randomly allocated to one of three conditions: the Scottish, Anglo-American, or special verdict conditions. In the former 2 systems, participants were asked ***“Which of the following verdicts do you think is most appropriate given the evidence. Remember, please only give a guilty verdict if you think the Crown has proven their case beyond reasonable doubt”***. In the special verdict condition, however, participants were asked ***“Please select whether you find each of the following statements proven or not proven, remember please only give a proven verdict for each of the statements if you think the Crown has proven their case beyond reasonable doubt. If you answer proven to 3 out of 3 statements, the judge will give a guilty verdict:”***. The three statements in the special condition were designed to correspond to the facts of a rape trial that juries are directed to consider:(1)The accused did penetrate the complainer (i.e., the victim);(2)The complainer (i.e., the victim) did not consent;(3)The accused knew that they did not have consent from the complainer (i.e., the victim).

For each of these statements, the participants were asked to find them proven or not proven. Participants were also asked how confident they were in the verdict they gave, how guilty they believed the accused to be, and—in an open-answer format—why they gave the verdicts they gave or made the special verdict decisions they made; these questions were asked regardless of the verdict system condition.

Before participants could reach a decision, participants were given a brief definition of what the verdict terms they would see mean or might mean (in the case they existed, i.e., the verdicts used in the special verdict system). ***“In a Scottish court, a proven verdict does not exist. If it did exist, it might be a special verdict. Likewise, in this trial, a not proven verdict will also be used as a special verdict. A special verdict is when a juror will decide whether each fact has been proven or not proven ‘beyond reasonable doubt’, and the judgement of the accused is left to the court’. This is instead of a general verdict, such as a guilty verdict, where the evidence has been enough to prove ‘beyond reasonable doubt’ that the accused person committed the crime. Likewise, a not guilty verdict is also a general verdict, which is given when the evidence has not been enough to prove ‘beyond reasonable doubt’ that the accused person committed the crime.”***

The perceptions of the accused and the complainer were measured using a series of 10-point response scales ranging from ‘not at all’ (scored as 0) to ‘Very’ (scored as 10). For example, ***“How credible do you perceive the accused’s testimony to be?’ and ‘How professional do you perceive the accused to be?”***. The post-trial questions were scored on the same scale and asked about the participants’ experience taking part in this study with questions such as ***“How confusing did you perceive the verdict-system to be?”*** and ***“How satisfied are you with your jury experience?”***

Three open-ended questions were also asked: ***“Is there anything else you would like us to know about the trial or your experience? Please write as much or as little as you like’***; ***‘Why did you give this verdict? Please explain:”***; and ***“Why did you make the decisions you made? Please explain:”***. These responses were analysed using thematic analysis based on Braun and Clarke (2022) [45] to extract key themes relevant to this study’s aims and research questions. These themes were helpful when reconnecting findings with the literature and the quantitative analysis within this paper. Themes were predominantly noticeable terms, ideas, or expressions mentioned by the participants [46]. Each participant’s responses (*n* = 180) were read and re-read to determine key words, phrases, or themes of significance to the current study’s aims. The data extracts, which were drawn from the transcripts, informed the narrative analysis included within the results section of this paper. Under each theme, a narrative was presented outlining the core messages which embodied the theme.

### 2.5. Procedure

Each participant first read an online information sheet before they gave their consent to participate. After consenting, they gave their demographic information. Participants were then asked to complete the AMMSA scale. After this, participants were presented with the mock trial. Watching the video footage took 68 min and provided mock jurors with a realistic (albeit shortened) experience of proceedings in court. Jurors were presented with instructions from the judge and evidence from the complainer, a specialist doctor, and the accused. Throughout the trial, participants were also presented with questions and statements from the prosecution and defence advocates. Once the trial had finished, participants were asked to give a verdict. The verdict options available to participants differed and were dependent on the condition they were randomly allocated to: Scottish; Anglo-American; or special verdict system.

After giving their verdict, participants were asked six questions to ensure that they had watched and understood the video footage. If they answered more than three of the questions incorrectly, their data were excluded from the analyses. Participants were then asked further questions about the case, such as their confidence in their verdict, and about their perceptions of the accused and the complainer. After this, participants were asked questions about their experience with the trial and of the survey. Once this section had been completed, participants were provided with the debrief and the contact details of the principal investigator for the study.

### 2.6. Data Preparation

To allow all the comparisons between the different verdict systems, all conviction verdicts (guilty and three not provens in the special verdict system) were re-classified to conviction, and all acquittal verdicts (not guilty, not proven and at least one not proven in the special verdict system) were classified as acquittal. The measure made up of convict and acquit, the dependent variable, was conviction.

## 3. Results

The first section of the results investigates the predictive effects that verdict system and rape myths have on the conviction rate (related to the main hypotheses). The second explores ancillary questions that provide important context to the first section, such as the relationship that rape myths (through the AMMSA score) have with measures related to how the accused and complainer were perceived. Also in the second section is an analysis of the effect that verdict system had on key measures such as verdict system confusion and verdict system confidence. At the end of this section, the qualitative analysis will be discussed.

### 3.1. Verdict Systems’ and Rape Myths’ Predictive Effects on Conviction Rates

First, a binary logistic regression showed that the chance of conviction did not differ significantly between each of the three verdict systems, with 28 convictions and 32 acquittals (total 60) in the Anglo-American verdict system, 23 convictions and 32 acquittals in the Scottish verdict system (total 55), and 26 convictions and 39 acquittals in the special verdict system (total 65) [χ^2^ (2) = 0.595, *p* = 0.74, R^2^ = 0.004]. This model tested the predictive ability of the verdict system without considering the AMMSA scores.

Second, a point-biserial correlation was established to determine the relationship between the AMMSA score and conviction without considering the effect of verdict system. There was a negative correlation between the AMMSA score and conviction, which was statistically significant (rpb = −0.204, *n* = 180, *p* = 0.003; this remained significant when outliers (7) were removed).

Note: it should also be mentioned here that the mean AMMSA score was not statistically different across all three verdict systems (Anglo-American verdict system = 67.98; Scottish verdict system = 69.73; special verdict system = 66.59) conditions [F(2, 177) = 0.200, *p* = 0.82], showing that rape myths were consistent across the conditions; this remained true when outliers were removed.

Next, it was important to test whether a model containing the predictor variables (verdict systems and the AMMSA score) could reliably predict conviction and acquittal verdicts. The interaction between verdict systems and the AMMSA scale was also included in the model. A Likelihood-Ratio Test between the model with and without interaction was significantly in favour of the model with interaction (LR Test χ^2^ (2) = 7.54, *p* = 0.023). The Anglo-American verdict system (guilty and not guilty) was selected as the reference category (as it is the most common system used in the world), meaning the special verdict system (proven and not proven) and the Scottish verdict system (guilty, not guilty and not proven) were compared with the Anglo-American verdict system.

First, the model correctly predicts 61.1% of the convictions (Convict and Acquit). Second, it was found that the Scottish verdict system and the Anglo-American verdict system did not differ significantly in relation to increasing the log odds of a conviction [χ^2^ (1) = 0.001, *p* = 0.97, *B* = −0.038, β = 0.963]. Third, it was found that the special verdict system did significantly differ from the Anglo-American verdict system, with higher log odds of a conviction being reached [χ^2^ (1) = 4.02, *p* = 0.045, B = 2.276, β = 9.39], meaning that for the special verdict system in comparison to the Anglo-American system, there was a 2.28 increase in the log odds of a conviction, given that all other variables in the model were held constant (including the AMMSA score). Fourth, it was found that the AMMSA score was not a significant predictor of the log odds for conviction [χ^2^ (1) = 0.205, *p* = 0.65, *B* = −0.004, β = 0.996].

Verdict system and the AMMSA score had a significant interaction when predicting convictions; see Figure 1 for a visual display of interaction. Those in the special verdict system who had an increase in rape myth score by one unit were significantly less likely to convict (−0.41 reduction in log odds) compared to those in the Anglo-American verdict system who also had an increase in rape myth score by one unit [χ^2^ (1) = 5.68, *p* = 0.02, *B* = −0.041, β = 0.959]. However, those in the Anglo-American verdict system, with an increase in rape myth score by one unit, were as likely to convict as those in the Scottish verdict system with an increase in rape myth score by one unit [χ^2^ (1) = 0.023, *p* = 0.88, *B* = −0.002, β = 0.998]. This seems to suggest that the relationship between the AMMSA score and conviction is stronger in the special verdict system when compared to the alternatives.

Interestingly, Figure 1 highlights that those low in the AMMSA score category in the special verdict (proven/not proven) system were more likely to convict than those who were low in the AMMSA score category in the other two verdict systems (Anglo-American—guilty/not guilty—and Scottish—guilty/not guilty/not proven). However, those high in the AMMSA score category in the special verdict system were more likely to acquit than those high in the AMMSA score category in the other two verdict systems (Anglo-American and Scottish).

In a post hoc analysis, the reference category was changed from the Anglo-American verdict system to the special verdict system. This was to allow comparisons between the Scottish and special verdict systems. Due to this change, the intercept increased to a log odds of 2.44 and became significant [χ^2^ (1) = 7.67, *p* = 0.01]. In the following analysis, only significant and/or novel findings (comparisons not made with the previous reference category) will be discussed.

First, those in the Scottish verdict system were not significantly more likely to convict when compared with the special verdict system [χ^2^ (1) = 3.780, *p* = 0.052, *B* = −2.31, β = 0.099]. Second, this analysis showed that jurors in the Scottish verdict system who had an increase in rape myth score by one unit were significantly more likely to convict (0.039 increase in log odds, given all other variables are held constant) than those in the special verdict system who also had an increase in rape myth score by on unit [χ^2^ (1) = 4.747, *p* = 0.03, *B* = 0.039, β = 1.040]. Third, and most interestingly, when the reference category changed, the AMMSA score became a significant predictor of convictions [χ^2^ (1) = 10.220, *p* < 0.001, *B* = −0.046, β = 0.96].

In relation to the main hypotheses, the results in this first section demonstrate the following: (1) the ability of the AMMSA score to predict convictions depends on what variable is used as the reference category in the model (this will be further discussed in the next section); (2) the special verdict system leads to higher convictions than the two and three-verdict systems when controlling for the AMMSA score; (3) when controlling for the AMMSA score, there was no significant difference in convictions in the two-verdict system when compared to the three-verdict system. Interestingly, though, the analysis showed that the relationship between verdict system and conviction depended on the AMMSA score, with those with a low AMMSA score being more likely to convict in the special verdict system than the Anglo-American or Scottish verdict systems, with the opposite pattern being true for those with a high AMMSA score.

### 3.2. Ancillary Analyses

Correlations between rape myths and perceptions of the accused and complainer.

Table 1, below, shows the mean and standard deviation for each of the variables included in these analyses. Likewise, the table shows how each of these measures correlate with the AMMSA score.

The data were deemed sufficiently parametric (only four outliers present) to conduct Pearson’s correlations between the AMMSA score and each of the other measures in the table. All correlations used a one-tailed hypothesis as it was expected that rape myths would be associated positively with favourable judgements of the accused and negatively with favourable judgements of the complainer.

The results show that the AMMSA score had a significant relationship with several variables related to the perception of the accused and victim. These highlight generally that the higher a person scores on the AMMSA score, the more favourably they perceive the accused and the less favourably they perceive the complainer. Interestingly, when variables with large and medium correlations with the AMMSA score were included in a model (Nagelkerke R^2^ = 75.3); χ^2^ (6) = 148.13; *p* < 0.001) to predict conviction, four variables were found to be significant predictor measures of conviction; the perceived credibility of accused (*B* = −0.28, *p* = 0.04); the perceived level of deception from accused (*B* = 0.45, *p* = 0.01); perceived level of deception from victim (*B* = −0.51, *p* = 0.047); perception of guilt surrounding the accused (*B* = 0.50, *p* = 0.046)—these effects remained significant when verdict system was included in the model. Overall, these suggest that the more favourably the jurors perceived the accused, the less likely they were to convict, and the more favourably they perceived the complainer, the more likely they were to acquit. 

### 3.3. Verdict System Differences

Additionally, analysis was conducted to test if the following measures differed across verdict systems: (1) confidence in own verdict; (2) perceived guilt of the accused; (3) how confident they were in their verdict system; (4) how confused they were by their verdict system; (5) how satisfied they were with their jury experience. Only one significant but small effect was found, with confusion with verdict system differing across the factor [F(2, 177) = 3.75, *p* = 0.03, η^2^ = 0.041]. Tukey’s post hoc comparisons showed that the special verdict system (M = 4.54) was seen to be significantly (*p* = 0.02) more confusing than the Anglo-American verdict system (M = 3.35). The three-verdict system (M = 3.85) did not significantly differ from any of the other conditions.

Finally, one-way between-participant ANOVAs were conducted to assess if each of the different key measures of perception of accused and complainer significantly differed across the verdict systems. The only significant difference emerged in relation to how reliable the accused was perceived to be [F(2, 177) = 3.188, *p* = 0.044, η^2^ = 0.035]. The mean of the special verdict system (M = 5.17) differed significantly (*p* = 0.036) from the mean of the Anglo-American verdict system (M = 4.05). No other significant effects were found.

### 3.4. Demographic Analysis

First, there was a significant correlation between age and the AMMSA score [r(180) = 0.19, *p* = 0.007], which was a small effect size and showed that older individuals were more likely to believe in rape myths. Second, a *t*-test between independent samples showed that males (M = 73.01) were more likely to score higher in rape myths than females (M = 63.77) [t (173) = 2.29, *p* = 0.1, d = 0.35].

### 3.5. Qualitative Analysis

The open-answer responses from the online survey were analysed by employing Thematic Analysis [45]. This process allowed the researcher to analyse each response to determine if themes have occurred within the data. This process was repeated for each participant’s response. Once all the responses had been analysed (*n* = 180), the researcher observed these to establish if patterns of themes occurred across all of the survey responses. The researcher then extracted data (quotations) from the transcriptions; these themes (findings) are discussed within the analysis below.

### 3.6. Rape Myths

Rape myths were evident within the qualitative narrative from the participants’ responses. These myths are expressed in parallel: one, in favour of the accused (perpetrator), and two, as a ‘counter rape myth’ in favour of the complainer. Rape myths have been depicted in the following quotations:

***“The 2 witness accounts were inconsistent. The expert witness proved that there had been penetrative sex, but lack of internal trauma makes it impossible to prove if it was rape”*** (Juror 79). The lack of physical evidence of internal trauma appears to be this juror’s justification for concluding that rape could not have occurred. This belief and misunderstanding of rape trials is depicted in the following statement:


**
*“It is clear that jury members will examine the case based on their preconceived ideas about rape which are on the whole prejudiced against the victim, mainly due to media portrayal of women in society. Rape trials would be better dealt with by teams of professionals rather than misinformed members of the public in my opinion”*
**
(Juror 144).

The significance of lay people presiding over rape trials is felt strongly by this participant. The juror’s views can be further emphasised in the comments made by the following two jurors. All jurors have watched the same video footage, but Jurors 12 and 32 have conflicting views on the significance of the reporting time. One juror appears to comprehend that post-rape, a complainer will be traumatised, and that complainers might not immediately call the police and report the incident:


**
*“Looking at the crown’s argument it is incredibly unlikely that the defence would call the police swiftly after the incident out of a situation of pure revenge. She was emotional and clearly distressed by the events that occurred and although the accused had a viable story with the events that occurred that night, the witness statements about the medical exam leaned towards a more violent act of rape over compassionate sex”*
**
(Juror 12).

In contrast, another juror interpreted the time delay as an opportunity for the complainer to have second thoughts about the incident ***“She had mixed feelings then doubted herself and blamed him”*** (Juror 32). The time delay for this participant appears to be the rationale for presuming that the complainer is lying about rape, and that she instead had consensual intercourse but has now had time to consider her actions and altered the version of events to a rationale that is more favourable to the complainer.

Interestingly, but not surprisingly, some jurors expressed views which counter commonly held rape myths, where the jurors are expressing their views contrarily to each other:


**
*“The complainant’s statement was backed by medical evidence, and false rape claims are very rare. I was not convinced that she lied, and no evidence was presented to support that making false revengeful allegations was in her character. The defendant, however, did have motive to lie, and did not ensure consent was given”*
**
(Juror 44).

This juror is aware of the low number of false rape claims that exists (around 3% of rape cases [47]), and it appears that their rationale for their returned verdict was primarily based on the assumption of false rape cases being low. The juror, however, also indicated that the medical evidence presented confirms that the complainer was raped. Whilst the juror offers a narrative that the accused (perpetrator) did lie, the juror does not provide any supporting evidence as to their rationale for this. The complainer’s perceived credibility is demonstrated in the response of another juror.


**
*“The victim had no good reason to invent a story about being raped to gain “revenge” on her former partner. Ms. Stewart was far more convincing in the dock and the injuries sustained helped to prove that the sex was not consensual”*
**
 (Juror 47).

This juror commented that there was no justification for the complainer to “invent a story”, and they believed that the story was not an act of “revenge”. The motives of the complainer, however, did come under scrutiny with one juror; they chose to protect the reputation of the accused, rather than the consider the prospect that the complainer has been subjected to rape and the lifelong implications this will have on her life:


**
*“I do not believe the Crown made a convincing enough case in this instance. The evidence provided could be explained reasonably by the accused, which made it clear to me that his guilt could not be proved beyond a reasonable doubt. Looking at the whole picture, I am left with questions about Ms. Smith’s motives in continuing to contact the accused and offering him wine on the occasion in question. It seems entirely possible that his rejection prompted her to fabricate a rape that never occurred. Equally, he could have raped her—but again there is no evidence that proves this beyond a reasonable doubt.*
**



**
*Ultimately, I have chosen the not guilty verdict because both versions of events are plausible, but believing the complainant’s version if it is not accurate could lock an innocent man away and ruin his life”*
**
(Juror 45).

Rape myths have been highlighted within this research and how jurors evaluate the evidence in court. Their acceptance of certain rape myths can be witnessed in the quotations of the following two jurors; ***“The complainer’s lack of concern, there was a lack of belief that she wasn’t consenting. There was a 40-min delay in phoning the police and remove herself from a web of lies”*** (Juror 147). This juror believes that a 40 min time delay is sufficient time to fabricate rape. The juror has not considered that the time delay is due to the rape occurring and the complainer being left traumatised and temporarily paralysed. Instead, the juror proposes that the complainer initially consented to intercourse and was now suggesting that she has been raped and that intercourse was not consensual, proposing the element of revenge.

Another participant’s response insinuates that if a complainer demonstrates any perceived suggestible behaviours, they could be perceived as implicit consent for intercourse: ***“The history given during the evidence was that the defendant was controlling the action from suggesting he move in with the accuser, to his suggestion that he move out of the flat, to his returning for the TV set. It is more than possible that he purposefully misread the “signals” from the accuser into thinking that it was consensual and allowing him to commit the sexual intercourse against her wishes”*** (Juror 173). This juror’s response is concerning as they believe that “intercourse against her wishes” is consensual. These statements and attitudes further highlight the challenges posed by rape myths and in rape trials. Intercourse without explicit consent is rape, but this juror did not comprehend this in their response.

### 3.7. Challenges Presented by the Criminal Justice System

It is imperative to consider each of the elements of the Scottish Jury System in parallel, as factors such as not proven, corroboration, and beyond reasonable doubt are intrinsically linked. The participants have demonstrated this in their responses; it is therefore difficult to consider each of the following themes separately, as they must be considered in tandem.

This paper has presented various critiques of the verdict system in Scotland. Interestingly, some of the mock jurors commented on the constraints within the criminal justice system. Acknowledging the difficulties in rape crime convictions and the challenges posed by “beyond reasonable doubt”, they have found similar constraints in relation to the verdicts available within Scot’s law (guilty, not guilty, and not proven) with jurors reflecting upon their rational for the decision-making process, ***“it makes me understand why rapes go so unreported. It really is a ‘he said she said’ situation and I hate it. There doesn’t seem like any way to have definitive proof”*** (Juror 7). This juror expresses their hatred towards the criminal justice system and the verdict system overall.

This response coincides with other jurors’ as they have stated the difficulties involved in rape cases: ***“I think he probably did it, but there is doubt which according to the judge means you have to acquit”*** (Juror 27); it is unclear, however, if this would change in a guilty or not guilty verdict system. Based on the participant’s response that the accused “probably did it”, a guilty verdict may have been deliberate and returned. This response resonated with another respondent: the very system that is supposed to protect victims of crime makes it difficult to prosecute, particularly in rape cases.


**
*“I feel the system makes it very difficult to prosecute crimes such as rape, especially when the accused admits to having consensual intercourse as often there won’t be overwhelming evidence that proves otherwise beyond reasonable doubt. These situations are thus often one word against another. Given that most women are unlikely to be fabricating crimes, it is women that will ultimately pay the price, and the rapist who will be acquitted and spared prosecution. I believe the system needs to change to support women who fall victim of these crimes”*
**
(Juror 37).

The challenge posed by this juror is the misinformation that the complainer confessed to having consensual intercourse. It is unclear why this juror believed this to be the case, but it further highlights the challenges faced with juror bias and the impact that these have on the complainer and the verdict delivered. The verdicts deliberated were discussed consistently throughout the analysis.

These views are conveyed by numerous jurors as they have commented that it is ***“very difficult to administer a guilty verdict, although I believe he was guilty”*** (Juror 43); ***“I now understand how hard it is to decide on guilty or innocence in this type of case”*** (Juror 68); ***“Ultimately, I believe the accuser, but it comes down to what can be proven beyond reasonable doubt”*** (Juror 116); and ***“The accuser was completely believable on their testimony alone I do not dispute their version of events. However, the rules provided by the court do not permit me to determine the defendant as guilty as I do not believe there is sufficient corroboration on one hand, or evidence of guilt on the other. So, I had to follow the rule of innocent until proven guilty. I do not believe guilty was proven”*** (Juror 80). These jurors have delivered a verdict based on the instructions of the judge, but from their responses, it is evident that they believe the testimony of the complainer and, if the law permitted, their verdict would be in favour of the complainer and not the accused (perpetrator).

As discussed previously in this paper, Scot’s law requires corroboration of evidence to prove beyond all reasonable doubt that a crime was committed. The difficulties posed by this aspect within Scot’s law are seen in various responses from jurors. ***“I believe the witness testimony, but I felt there was nothing else to go on and without more evidence against the accused, because the physical evidence was disputed, that I had to choose ‘not proven’***” (Juror 94). As with many jurors, Juror (129) detailed their confusion regarding decision-making when providing a verdict, particularly in relation to the not proven verdict: ***“Have always found the not proven verdict in Scotland leads to confusion and maybe even influences some people’s decision so that maybe they see it as an easy way out of committing to a decision one way or the other therefore maybe not providing a resolution for the victim”***. ***“I can understand now how hard it can be to prosecute when there isn’t enough corroboration which is only use in Scottish Law and see how difficult it is for victims to get the results and it is so frustrating”*** (Juror 172).

Jurors have consistently depicted their frustration towards the Scottish judicial process and the impact of requiring corroboration, particularly in rape case trials. The criminal justice system in Scotland has been discussed extensively throughout the responses detailed by the respondents (participants), and for the participants that did believe the testimony, they were restricted in the verdict that they could deliver: ***“The accuser was completely believable on their testimony alone I do not dispute their version of events. However, the rules provided by the court do not permit me to determine the defendant as guilty as I do not believe there is sufficient corroboration on one hand, or evidence of guilt on the other. So, I had to follow the rule of innocent until proven guilty. I do not believe guilt was proven”*** (Juror 80). This response was previously discussed in relation to the verdict system; like many responses from participants, they have been found across more than one theme. Similarly, responses discussed within this paper are evidenced across and within the themes. The following participant felt that ***“It would have been useful to hear more about the psychological impact on the victim from* e.g., *police evidence, Doctor and family/friends. This might have been another piece of corroboration”*** (Juror 66). According to this juror, the introduction of a victim impact statement would have persuaded their decision if they had this as corroborating evidence.

Recent changes to Scot’s Law, however, allow for the introduction of ‘distress’ that has been observed by a third party (de recanti) to corroborate the complainer’s narrative that rape has occurred [47]. The introduction of distress in sexual assault and rape cases will be pivotal and a very welcome development within Scotland. The court can now draw upon the complainer’s account of distress if this has been witnessed by an independent person on the account that she or he states that they have been raped. The previous guidance surrounding corroboration was often confusing for juries in rape trials. This advancement in guidance is in part to provide clearer guidance surrounding corroboration, thus in turn avoiding misdirection from judges, which can ultimately lead to appeals and acquittals and the failure of the survivor by the criminal justice system [47].

Rape has a long-lasting psychological impact on the victim; this juror recognises this in their response, but they fail to recognise the effect on the victim of having this detailed in court. The criminal justice process and the behaviours of defendant lawyers have been called into question in Scotland recently [48], and similar reviews may be enabled by high court transcripts being made available to complainers of rape in Scotland [32]. Whilst the behaviour of the lawyers was not discussed within this research, the credibility of the evidence was discussed: ***“I thought it was very convincing. I was torn but struggle to see how this sort of trial can be proved when both stories are credible”*** (Juror 9). This response unearths the narrative of the challenges that are faced by jurors in rape trials within Scotland and the apparent need for more decisive evidence to prove guilt or innocence, especially in rape trials.

## 4. Discussion

There were three main hypotheses tested in the current study. The first was ‘(1) the AMMSA score will predict convictions, the higher the rape myth score, the fewer convictions there will be’. This hypothesis cannot be rejected and needs to be further qualified. A point-biserial correlation initially highlighted a weak but significant correlation between convictions and the AMMSA score, with those with higher AMMSA scores being less likely to convict. However, when included in a binary logistic model of interaction with the verdict system, the association became non-significant. The association was significant though when the reference category changed from the Anglo-American to the special verdict system. Therefore, the analyses suggests that the AMMSA score predicted convictions most strongly when participants passed their verdict under the special verdict system, with those less affected by rape myths being more likely to convict in the special verdict system than the other two systems. This suggests that the effect of rape myth score on conviction is mainly due to a strong association for the special verdict system rather than all three verdict systems. Further, Figure 1 and the estimated coefficients indicate a tendency of individuals in both the Anglo-American and the Scottish verdict systems to be more likely to acquit when they were highly affected by rape myths. Furthermore, rape myths did influence how convictions were reached, but the relationship was much stronger in the special verdict system.

The qualitive analysis also suggested that rape myths were influencing the decision-making of jurors. Many jurors provided responses to open-answer questions in which they perpetuated rape myths in their rationales for returning their verdicts. Many participants also indicated their awareness of rape myths, with some also stating that they understood how difficult it was to come to conclusions in these case types given the ‘he said, she said’ nature of the evidence. It is apparent that some jury members will examine the case based on their preconceived concepts (rape myths) about rape cases, which are primarily prejudiced against the victim. The media’s portrayal of women in society and increases in misogynistic attitudes are conveyed across various social media platforms (such as TikTok, Instagram, and Facebook [49,50,51]).

The AMMSA score was also found to correlate with several variables (such as perception of guilt of the accused and perceived level of deception by the complainer)—when collapsing the verdict system condition—and some of these variables (e.g., the perception of deception by the accused) also predicted conviction. This suggests that although rape myths had the strongest effect on convictions in the special verdict system, they may have been influencing how jurors viewed both the accused and complainer in all conditions, which then influenced the verdicts that they reached. Generally, jurors with highly rape myth-based views were likely to perceive the accused more favourably and the complainer more negatively (see Table 1), and jurors who viewed the accused more favourably were less likely to convict and those who viewed the complainer more favourably were more likely to convict.

This finding is relatively novel, as it is the first piece of empirical research to highlight that rape myths influence how the legal actors (accused and complainer) are perceived, which then influences convictions in two- and three-verdict and special verdict systems. Previous research has hinted that this may be the case, however. For example, research from Chalmers, Leverick, and Munro (2021) [31] showed that in mock jury deliberations, there was “considerable evidence of the expression of problematic attitudes towards rape complainers” (p. 226). For example, they also found that some jurors believed that the complainer was deceiving the court. They suggested that she could not have been raped as she did not resist her attacker, did not have injuries, and because she delayed reporting the rape. Notably, each of these beliefs was also demonstrated in open-answer responses in the current study.

It is important to note that the current study used the same mock trial material as Chalmers et al. (2021) [31]. Research using similar materials has been conducted in other jurisdictions (with a two-verdict system), showing that rape myths influence the way in which the accused and complainer are perceived [26,27,52,53].

The second hypothesis was that ‘(2) When controlling for AMMSA score, the special verdict system (Proven and Not Proven) will lead to more convictions than either of the general verdict systems (Anglo-American, two-verdict system and Scottish, three-verdict system)’. When controlling for the effects of rape myths in a binary logistic regression, it was found that convictions were much more likely in the Special verdict system of proven and not proven when compared to the Anglo-American verdict system; the difference was also approaching significance when comparing the special and Scottish verdict systems.

One reasoning for this is that the special verdict system, when controlling for the effects of rape myths, may cause jurors to focus on the facts of the trial [9,38]. They were also told this in the Anglo-American and Scottish verdict system at the end of the trial. However, it was not made explicit during the process of reaching their decision, as jurors were asked to judge the trial more generally to give a general verdict. Therefore, the explicit request to answer three concrete questions in the special verdict system regarding the trial rather than the one abstract question focused on guilt (used in both of the general verdict systems) may have made the decision more manageable [54].

Research has shown that a concrete/abstract paradox exists “whereby when a particular problem elicits opposing responses depending on whether it is framed abstractly or concretely” [54] (p. 3). For example, people often answer no when asked if people are fully responsible for their actions in an already determined universe (abstract decision) but would answer yes if asked a person is responsible for murder [54] (p. 3), Further, people are more likely to donate money to a specific person rather than to an abstract sense of group suffering [55,56]. One reasoning for this is that concrete terms (e.g., proven or not proven) may promote empathic responses (‘what if that were me?’ ‘What if that was my wife, friend or daughter?’), whereas abstract terms (e.g., ‘guilt’ or ‘innocence’) may cause a feeling of detachedness in the jurors. Interestingly, concrete terms even seem to activate different brain areas compared to abstract terms [57]. Therefore, concrete terms of proven and not proven (when controlling for rape myths) may increase empathy for the complainer, thus increasing convictions. However, there was no evidence that the varying verdict systems caused people to be more empathetic towards the complainer.

An alternative explanation, therefore, may be that by separating a general decision of guilty or not guilty into three manageable decisions of proven or not proven, the decision-makers have more cognitive capacity to evaluate the evidence and think about whether or not each of the facts has been proven (i.e., they have a higher elaboration likelihood [41]). However, to fully explore whether the proven and not proven system is increasing convictions through rational or more emotional processes, further investigation would be required that explicitly measures the emotionality of the decision process [58]. This result may also suggest that in other crime types, where the effects of rape myth would not be present, a special verdict system may increase convictions, as effects on convictions may differ depending on what bias is being attenuated (i.e., pro-defence due to rape myths vs. pro-prosecution) or who the system is causing the jurors to relate to (i.e., the complainer or the accused).

Interestingly, though, there was also a significant interaction between verdict system and rape myths in relation to convictions. This highlights that the influence that rape myths have on convictions depends on the verdict system, with jurors with highly rape myth-based attitudes being more likely to acquit in the special verdict system when compared to jurors with highly rape myth-based attitudes in the Anglo-American and Scottish verdict systems. However, those less affected by rape myths in the special verdict system were more likely to convict than those in the other verdict systems. One reasoning for this may be that when three decisions are available, rather than one, there is a higher chance of rape myths influencing at least one of the decisions. For example, jurors with highly rape myth-based attitudes may think that “I believe that they did not consent, but I do not believe he was aware of this, so will give not proven for the final decision”, thus decreasing the conviction rate, or alternatively, when less affected by rape myths, this view will influence all three of these decisions.

This result may also suggest that the special verdict system may cause jurors to focus on the key components of a charge (similarly, the indictment, the route to verdict, or the discussion in the deliberation room may do the same), causing their beliefs to influence each of the respective components (Did the accused know they did not have consent?). Now, where the juror falls on the rape myth scale it may influence their chances of convicting, with those low on the scale being more likely to convict than those higher on the scale [31,53,59]. This may provide evidence—when taken in combination with the previous result of when rape myths were controlled for—for deselecting jurors based on rape myths and thus controlling for them in the courtroom; this is discussed further in the recommendations section.

The third hypothesis was that (3) ‘When controlling for AMMSA score, there will be more convictions in the Anglo-American system when compared to the Scottish verdict system’. This hypothesis can be rejected as when controlling for rape myths, there was no difference in the conviction rates between the Anglo-American and Scottish verdict systems. Rather, the relationship between rape myths and conviction became stronger in the special verdict system, as already discussed above.

### 4.1. Limitations

A key limitation of this study was that, despite our ecologically valid and realistic materials, the participants did not engage in deliberations and knew that their decisions had no consequences [59,60,61]. Both of these issues have been raised previously [59,61] and limit the ability of findings to be generalised to the courtroom.

However, due to legal constraints, researchers do not have access to real juries, whose decisions have real consequences. Likewise, due to ethical constraints, it may not be wise to deceive mock jurors into thinking that their decisions do have real consequences, as real jurors with real consequences can suffer negative effects such as trauma and poor mental health [62]. The lack of access to real juries also inhibits the ability of researchers to detail the full course of how cases of rape and sexual assault proceed through the criminal justice system [63]. We share the opinion of Krauss and Lieberman (2017) [60] that the best method to rectify this is by trying to triangulate data using different methods and analysis. Indeed, this is why the authors have also engaged with legal stakeholders [38] and conducted meta-analyses on the subject [23].

Deliberations, although costly in both financial and time senses, can be included within research studies [60]. However, due to the suggestions for reform in the “Victims, Witnesses and Justice Reform” (Scotland) Bill (2023) [8], the future of the Scottish jury will likely look very different. Therefore, the current study hoped to focus on one factor most pertinent to rape. Once the jury-size and majority-size reforms have been agreed in parliament, this study will be replicated with a jury.

A second limitation is that the scales used to measure rape myths (such as the AMMSA scale) have been critiqued for their use of language [26]. For example, phrases such as “a lot of women strongly complain about sexual infringements for no real reason, just to appear emancipated” may confuse participants [26] (p. 258). Further, the scales may cause a social desirability bias, where mock jurors respond in a way that is socially desirable rather than what they believe [26]. Whether it is complexity, a lack of understanding, and/or a social desirability bias, it is likely that only a small number of rape myths are captured by these scales. Therefore, we tried to address this by also asking participants how they perceived the accused and complainer (to assess if these perceptions aligned with what we would expect when the juror held rape myths) and analysing qualitative data.

### 4.2. Recommendations

The qualitative analysis suggested that there was frustration with the current system and a thirst for reform. The main recommendations of reform that can be made from this paper are two-fold. Based on this and related research [31], it is evident that rape myths are colouring how jurors view the accused and complainer and thus influencing the fairness of the trial. Therefore, one recommendation is to deselect jurors from rape trials based on rape myths, using scales such as the AMMSA scale [14]. Although the scales are not perfect, they would help to remove from consideration prospective jurors with problematic beliefs regarding rape and sexual assault [12]. Currently jurors are summoned via post and respond online. During the online phase, jurors could answer questions relating to rape myths, which could signal to the court whether they are appropriate for a rape trial or whether these jurors would be better directed to be selected for another trial type (e.g., murder or physical assault); similar to how prospective jurors fill out jury validation questionnaires in the USA. Then the final jury could be selected from a pool of prospective jurors with reasonable beliefs around rape (for instance, by setting a threshold on the scale). This would be a rather effective method for the court to apply since rape myth questionnaires are freely available and take only 5–10 min to complete.

Once jurors with rape myths have been deselected, we recommend that the current verdict system be replaced with the special verdict system of proven and Not proven. There are several justifications for our reasoning. First, after controlling for rape myths, the special verdict system is the first system ever to be shown to increase conviction rates when compared to the Anglo-American and Scottish verdict systems. Ormston et al. (2019) [1] and Hope et al. (2008) [2] found that convictions did not significantly differ between the Anglo-American and Scottish verdict systems in rape and sexual assault trials. The special verdict system might also satisfy legal professionals who have expressed a preference for a proven and not proven, binary verdict system, and campaigners against who want reform from the current Scottish, three-verdict system (Curley et al., 2022) [3].

Second, this recommendation may inform the “Victims, Witnesses and Justice Reform” (Scotland) Bill (2023) [8], as juries could reach special verdicts of proven and not proven on a number of different aspects of a trial, and the judge could use this information (similar to how the researchers did in this study) to make a general verdict of either guilty or not guilty. This would mean that in rape trials, the accused is still judged by a jury of their peers, but the ultimate decision is left to the judge (informed by a prior decision made by a jury). This would mean that a judge may reject the juries’ decisions if they believe that the decision is not legally warranted (i.e., bias/rape myths have influenced their decision). The recommendation might also contribute to wider discussions regarding policy associated with rape justice in the UK, such as regarding reform of rules associated with the UK’s Criminal Injuries Compensation Scheme in cases of rape [64].

Third, a proven and not proven system would allow for more transparency within the criminal justice system (something desired by the Scottish Government and by complainers of rape [32]), highlighting to the court and the public exactly what the jury believed to be proven and what the jury believed to be not proven. There has been critique in the press [65] surrounding jury deliberations and the manner in which they are conducted; therefore, more transparency through a special verdict system of proven and not proven may help to regain public confidence in the jury system. Considerable inequalities in the outcomes for women of different ages and ethnicities regarding rape and sexual assault trials [66] might also be addressed by increased transparency.

### 4.3. Conclusions

In conclusion, there are three main findings from this study. First, that rape myths may influence how the accused and complainer are perceived, which then influence conviction rates. Second, when controlling for the effects of rape myths, the special verdict system leads to more convictions than the two- and three-verdict systems. Third, rape myths have their strongest effect on convictions in the special verdict system, which may also have implications for deliberations and indictments. The qualitative data demonstrated significant rape myths within the participants’ responses. The commentary from the participants also captures several negative emotions surrounding the Scottish judicial process and the sense that the solutions are not easy to come by given the ‘hidden’ nature of sexual assault and rape.

Nevertheless, there are several limitations of this study. For example, this study did not include jury deliberations, and traditional rape myth measures may not capture all types of rape myths. However, we hope to address these limitations in a future study replicating this study with mock jury deliberations. Recommendations for change based on the content of this paper include jury selection and a change to the special verdict system of proven and not proven, as we believe this may increase conviction rates, work within the “Victims, Witnesses and Justice Reform” (Scotland) Bill (2023) [8], and regain public confidence in the jury system by increasing transparency in relation to jury decision-making.

## Figures and Tables

**Figure 1 behavsci-14-00619-f001:**
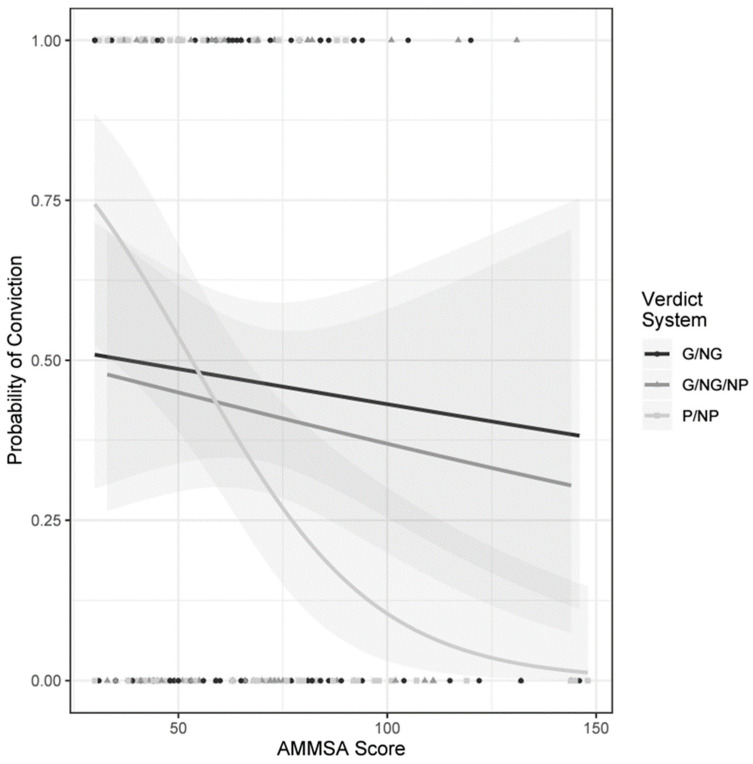
Illustration of the estimated interaction between verdict system and the AMMSA score on conviction. Each curve describes the probability for a conviction as a function of rape myth score for each of the verdict systems (Anglo-American—G/NG (circle); Scottish—G/NG/NP (triangle); Special—P/NP (square)). The shaded ribbons indicate ± 1SE of estimates. See text for details. Source: Author’s own creation.

**Table 1 behavsci-14-00619-t001:** Correlation between key measures of perception of accused and complainer and rape myths. As a guide, the strength interpretation of correlation has been assigned a label according to the following: small (0.01 to 0.29), medium (0.3 to 0.49), and large (0.5 to 1).

*Variable*	*N*	*M*	*SD*	*R*	*Strength*
AMMSA Score (Rape Myth Measure)	180	68.01	26.98	N/A	N/A
Perceived Credibility of the Accused’s Testimony	180	4.74	2.49	0.31 **	Medium
Perceived Reliability of the Accused’s Testimony	180	4.58	2.54	0.28 **	Small
Perceived Professionalism of the Accused	180	6.39	1.98	0.02	Small
Perceived Level of Education of the Accused	180	6.91	1.44	−0.01	Small
Perceived Likeability of the Accused	180	5.30	2.08	0.24 **	Small
Perceived Trustworthiness of the Accused	180	4.48	2.24	0.26 **	Small
Perceived Confidence of the Accused in their own Testimony	180	5.88	2.31	0.22	Small
Perceived Accuracy in the Accused’s Recall	180	4.73	2.4	−0.27 **	Small
Perceived Level of Deception from Accused	180	6.16	2.58	−0.30 **	Medium
Perceived Level of Ability to Convince from Accused	180	4.92	2.58	0.21	Small
Perceived Ability to Change Opinion of the Reliability of the Accused	180	4.07	2.16	0.17	Small
Perceived Trustworthiness of the Complainer	180	7.46	2.00	−0.29 **	Small
Perceived Credibility of the Complainer’s Testimony	180	7.8	1.92	−0.28 **	Small
Perceived Reliability of the Complainer’s Testimony	180	7.48	1.99	−0.25 **	Small
Perceived Professionalism of the Complainer	180	7.45	1.53	−0.19 **	Small
Perceived Level of Education of the Complainer	180	7.35	1.36	−0.093	Small
Perceived Likeability of the Complainer	180	7.07	1.7	−0.15 *	Small
Perceived Trustworthiness of the Complainer	180	7.46	2.00	−0.29 **	Small
Perceived Confidence of the Complainer in their own Testimony	180	7.42	1.95	−0.14 *	Small
Perceived Accuracy in the Complainer’s Recall	180	7.74	1.79	−0.36 **	Medium
Perceived Level of Deception from Complainer	180	2.13	2.13	0.50 **	Large
Perceived Level of Ability to Convince from Complainer	180	7.95	1.89	−0.36 **	Medium
Perceived Ability to Change Opinion of the Reliability of the Complainer	180	3.75	2.19	0.19 **	Small
Perception of Guilty surrounding the Accused	180	6.95	2.27	−0.35 **	Medium
Verdict Confidence	180	7.2	1.95	0.07	Small

Note: After Bonferroni’s correction, adjusted *p*-values were significant at * = *p* < 0.002 and ** = *p* < 0.001. Source: Author’s own creation.

## Data Availability

Data will be made available via the Open University library Open Research Data Online (ORDO) repository (link to be made available for publication).

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
