# Peer review of "Rape Myths and Verdict Systems: What Is Influencing Conviction Rates in Rape Trials in Scotland?"

_behavsci, 2024, doi:10.3390/bs14070619_

Round 1

Reviewer 1 Report

Comments and Suggestions for Authors

This is an excellent empirical investigation of juror decision-making for any jurisdiction given the high degree of ecological validity with the use of a 68 minute video trial simulation. However, what makes this study especially important is the investigation of different verdict systems. The results are critical in understanding the role of rape myth acceptance in verdicts, and how jurors with lower rape myth acceptance in the special verdict system are more likely to convict. The qualitative data is also very useful and an often-excluded component in jury research. The implications for practice are clear and compelling. Future research should include a deliberation component (as recommended by the authors) and test to see if the same patterns of results are present.

There are a few minor typographical errors (e.g., inconsistent spacing and use of italics in statistics), but the editor should be able to polish those. Those are the only recommended revisions.

Author Response

Comments 1: This is an excellent empirical investigation of juror decision-making for any jurisdiction given the high degree of ecological validity with the use of a 68 minute video trial simulation. However, what makes this study especially important is the investigation of different verdict systems. The results are critical in understanding the role of rape myth acceptance in verdicts, and how jurors with lower rape myth acceptance in the special verdict system are more likely to convict. The qualitative data is also very useful and an often-excluded component in jury research. The implications for practice are clear and compelling. Future research should include a deliberation component (as recommended by the authors) and test to see if the same patterns of results are present.

Response 1: We would like to thank the reviewer for their positive consideration of our work. We’re glad they agree on the quality and importance of the work and its future direction.

Comments 2: There are a few minor typographical errors (e.g., inconsistent spacing and use of italics in statistics), but the editor should be able to polish those. Those are the only recommended revisions.

Response 2: Thank you for pointing out these minor errors. We have made another effort to correct all of these we were able to find.

Reviewer 2 Report

Comments and Suggestions for Authors

This is an important, interesting and original contribution to the literature on the Scottish legal system and in particular, the verdict system, and jury trials in relation to cases of sexual violence. 

The paper is based on a well designed piece of research and the presentation is good. I would suggest some amendments to improve readability and accessibility. Also some sections/statements need checking. 

When setting out the explanations/definitions in the study - e.g. on line 272 - placing these in a box or different font, would make the text easier to read. This also applies to survey questions and importantly quotes from the research participants - these are not indicated clearly at all and should be - see line 493.

The presentation of findings is difficult to follow in places - this is because this is quite a complex analysis but I wonder whether there is some room for making the presentation more accessible - maybe moving some of the stats into footnotes, using the hypotheses as a reference for the findings... Can you check if the findings presented in lines 447-449 are correct - they seem counterintuitive and the opposite is stated in lines 702/703. 

The sections on the demographic factors was briefed - any correlations between ethnicity, level of education, etc?

Indicating juror comments clearly is important for the readability of section 3.6. In that discussion you use the term misogyny - please define it - I am not convinced by labelling the quotes presented as misogyny. I am not sure that the quote suggests that the juror believes that 'intercourse against her wishes is consensual' - please check. 

Line 573, incomplete sentence.

In some places, details of the mock trial are referred to which are not previously explained or it is unclear what the comments relate to - e.g. in line 588, there is reference to the accused admitting intercourse and in line 594, the reference is to misinformation and the complainer admitting?? I wonder whether including some details on the trial and the evidence presented would help - maybe as an appendix or linked data?

The quote from Juror 80 is used twice lines 605 and lines 630.

The discussion is sound but the reference to A Tate not necessary here - rape myths are fairly persistent and definitely not new - not sure this adds anything. 

Comments on the Quality of English Language

English Language is fine, just one incomplete sentence and a little more clarity and accessibility in the presentation of findings required. 

Author Response

Comments 1: This is an important, interesting and original contribution to the literature on the Scottish legal system and in particular, the verdict system, and jury trials in relation to cases of sexual violence.

Response 1: We would like to thank the reviewer for their positive consideration of our work and the excellent, detailed comments they made which will serve to improve the work.

Comments 2: When setting out the explanations/definitions in the study - e.g. on line 272 - placing these in a box or different font, would make the text easier to read. This also applies to survey questions and importantly quotes from the research participants - these are not indicated clearly at all and should be - see line 493.

Response 2: We are not sure if the editor / journal has a preferred format for displaying quotations and other significant text. In the meanwhile, we have made an effort to differentiate this text from the rest of the paper by using bold and italics. You can see examples of this on line 320, 551 and elsewhere.

Comments 3: The presentation of findings is difficult to follow in places - this is because this is quite a complex analysis but I wonder whether there is some room for making the presentation more accessible - maybe moving some of the stats into footnotes, using the hypotheses as a reference for the findings... Can you check if the findings presented in lines 447-449 are correct - they seem counterintuitive and the opposite is stated in lines 702/703.

Response 3: We believe the difficulty in following the findings may have been in part due to the omission of a design and participants subsection in the Materials and Methods section. We have added these to improve the clarity of the results section. We are not sure if footnotes are acceptable in the format of the journal so decided not to add these at this stage. Regarding findings presented in what are now lines 512 - 514 – thank you for pointing out this error in our stated direction. This should read “…the more favourably the jurors perceived the accused, the *less* likely they were to convict…”. We have highlighted this change in text.

Comments 4: The sections on the demographic factors was briefed - any correlations between ethnicity, level of education, etc?

Response 4: The majority of the demographic information we collected and reported were used to help us screen participants to include those who met the requirements for jury service in Scotland.  We have added a participants section which more clearly reports this information. Most relevant to this comment, almost 94% of the sample were White Scottish/British so any comparison between these and the small number of participants of other ethnicities in terms of AMMSA score would not make for strong conclusions. We also did not ask participants for their level of education.

Comments 5: Indicating juror comments clearly is important for the readability of section 3.6. In that discussion you use the term misogyny - please define it - I am not convinced by labelling the quotes presented as misogyny. I am not sure that the quote suggests that the juror believes that 'intercourse against her wishes is consensual' - please check.

Response 5: According to your comment 2 we have more clearly highlighted juror comments We have also removed references to misogyny.

Comments 6: Line 573, incomplete sentence.

Response 6: Thank you for pointing this out. We have amended this sentence, so it is no longer incomplete.

Comments 7: In some places, details of the mock trial are referred to which are not previously explained or it is unclear what the comments relate to - e.g. in line 588, there is reference to the accused admitting intercourse and in line 594, the reference is to misinformation and the complainer admitting?? I wonder whether including some details on the trial and the evidence presented would help - maybe as an appendix or linked data?

Response 7: Thank you for this comment. This comment refers to statements made by our participants in open-answer segments of our survey about the contents of the trial. While we agree that including some detail might be helpful to give some context to participant comments, we do not have access to a transcript of the trial as the video was provided by a third party. It is possible some participants may have misinterpreted, misremembered or otherwise not fully understood parts of the trial which might explain the identified discrepancies.

Comments 8: The quote from Juror 80 is used twice lines 605 and lines 630.

Response 8: After the second use of this quote it is acknowledged in the text that the quote was referred to twice as it fit across multiple relevant themes. We have decided to keep both now for this reason.

Comments 9: The discussion is sound but the reference to A Tate not necessary here - rape myths are fairly persistent and definitely not new - not sure this adds anything

Response 9: We agree that rape myths are persistent and pervasive without making reference to a particular individual and have deleted referred to Tate.

Reviewer 3 Report

Comments and Suggestions for Authors

This manuscript considers the 3-verdict system used in Scotland for rape trials and compares this system to two other verdict systems used in other locales for the relative conviction rate in cases of rape/sexual assault.  In addition, it measures the relevances of juror adherence to rape myths to verdicts within each of the three systems, with the main goal of understanding whether there are improvements to the rate of conviction that can be made within Scotland's system. The authors do well in explaining the background of the 3-verdict system, its potential weaknesses with respect to convictions in rape trials, and its relative effectiveness in conviction rates as compared to the other two systems under conditions related to juror adherence to rape myths. Juror questionnaires to assess rape myth adherence were administered, and filmed rape trials scenarios featuring legally trained personnel were shown to individuals to determine the verdicts that would be rendered under different systems.

This manuscript is nicely polished and in excellent condition in terms of background, structure, study design and description, findings and discussion.  It is thorough and makes a definitive contribution to the field, with concrete suggestions for application to the legal system at the end of the paper.  The only relative weakness detected is in the number and range of citations, which could certainly be somewhat more broad and numerous.  One are in particular that could use additional citation is in demonstrating the need for ways to address the conviction rate for rape/sexual assault cases in the first place.  There should be some conviction rate statistics to compare with other crimes within the same judicial system (Scotland)? The authors are clear in noting that the government's position in other ways is not well supported with research, and this paper is offered as a partial remedy to address this lack of scholarship.

Explanations of specific juror adherence to rape myths is a well-developed and section that is a welcome addition to this excellent paper, and the specific recommendations for changes to the legal system at the end are clear and meaningfully connected to the analysis.

Author Response

Comments 1: This manuscript is nicely polished and in excellent condition in terms of background, structure, study design and description, findings and discussion.  It is thorough and makes a definitive contribution to the field, with concrete suggestions for application to the legal system at the end of the paper.

Response 1: We would like to thank the reviewer for their positive consideration of our work. We’re glad they agree on the quality and importance of the work and its future direction.

Comments 2: The only relative weakness detected is in the number and range of citations, which could certainly be somewhat more broad and numerous.  One are in particular that could use additional citation is in demonstrating the need for ways to address the conviction rate for rape/sexual assault cases in the first place.  There should be some conviction rate statistics to compare with other crimes within the same judicial system (Scotland)?

Response 2: Thank you for this comment. We have made an effort to broaden our sources, especially regarding demonstrations of the need to address conviction rates for rape and sexual assault cases, particularly in Scotland. We have included 9 additional sources and made reference to them in the text (these are sources 58 - 66). These additional citations and associated references can be found in text throughout the introduction and discussion and respectively in the references section. Examples can be found on lines 125, 147, 154, 163, 171, 861 and 927. Regarding the conviction rate statistics to compare with other crimes within Scotland, our current source [6] provides these statistics, though we have updated the source to a more recent version.